# Genomic Characterization of Canis Familiaris Papillomavirus Type 25, a Novel Papillomavirus Associated with a Viral Plaque from the Pinna of a Dog

**DOI:** 10.3390/ani13111859

**Published:** 2023-06-02

**Authors:** John S. Munday, Kristene Gedye, Matthew A. Knox, Lynne Robinson, Xiaoxiao Lin

**Affiliations:** 1Pathobiology School of Veterinary Science, Massey University, Palmerston North 4442, New Zealand; 2Molecular Epidemiology Laboratory, School of Veterinary Science, Massey University, Palmerston North 4442, New Zealand; k.gedye@massey.ac.nz (K.G.); m.knox@massey.ac.nz (M.A.K.); 3Robinsons Veterinary Clinic, Feilding 4702, New Zealand; 4Massey Genome Service, Massey University, Palmerston North 4442, New Zealand; x.x.lin@massey.ac.nz

**Keywords:** dog, papillomavirus, viral plaque, pigmented plaque, CPV, canine papillomavirus, CPV25, neoplasia, skin

## Abstract

**Simple Summary:**

Papillomaviruses (PV) are known to cause a variety of skin lesions in dogs, including hyperplastic papillomas (warts) and viral plaques. There are currently 24 different PV types that are fully characterized in dogs. These PV types are divided into three genera. The *Chipapillomavirus* genus contains the largest number of canine papillomavirus (CPV) types. The *Chipapillomaviruses* are recognized to cause viral plaques in dogs. These plaques are usually multiple in nature and develop most frequently on the ventrum. In the present case, a dog developed a cluster of viral plaques that was confined to the concave surface of one pinna. Papillomaviral DNA was amplified and found to be a novel CPV type. As the novel type was the 25th PV identified in domestic dogs, the PV was designated CPV25. Many of the cells within the plaque showed PV-induced cell changes, suggesting that CPV25 was the cause of the plaques. As this is the first report of CPV25, this CPV type may rarely infect dogs and may be, at most, an uncommon cause of viral plaques in dogs. While further cases are required, CPV25 may cause canine viral plaques that develop at unusual locations and have atypical histological features.

**Abstract:**

A 14-year-old West Highland White terrier dog developed multiple raised plaques that were confined to the concave surface of the right pinna. Histology allowed a diagnosis of viral plaque, although the lesions contained some unusual microscopic features. A papillomaviral (PV) DNA sequence was amplified from the plaque using consensus PCR primers. The amplified sequence was used as a template to design ‘outward facing’ PCR primers, which allowed amplification of the complete PV DNA sequence. The sequence was 7778 bp and was predicted to code for five early genes and two late genes. The *ORF L1* showed the highest (83.9%) similarity to CPV15, and phylogenetic analysis revealed the novel PV clustered with the species 3 *ChiPVs*. The novel PV was designated as canine papillomavirus (CPV) type 25. As CPV25 was not previously detected in a canine viral plaque, this PV type may be a rare cause of skin disease in dogs. However, as plaques that remain confined to the pinna were not previously reported in dogs, it is possible that CPV25 could be more common in plaques from this area of skin. The findings from this case expand the number of PV types that cause disease in dogs. Evidence from this case suggests that, compared to the other canine *ChiPV* types, infection by CPV25 results in viral plaques in atypical locations with unusual histological features.

## 1. Introduction

Papillomaviruses (PVs) are a common cause of hyperplastic, and an uncommon cause of neoplastic, disease in dogs [1]. Papillomaviruses express four or five early (E) genes and two late (L) genes. The nucleotide sequence of the *ORF L1* is used to classify PVs with greater than 60% similarity, usually within the same genus; PVs with greater than 70% similarity within the same species; and PVs with greater than 90% similarity the same type [2]. There are currently 24 canine papillomavirus (CPV) types, which are divided into 3 genera [3]. The *ChiPV* genus contains the largest number of CPV types, with 14 *ChiPV* types subdivided into 3 species. PVs within genera typically cause the same lesions in closely-related host species [2].

Viral plaques (also called pigmented viral plaques) are uncommon skin lesions of dogs. They typically appear as multiple small and slightly raised dark plaques over the ventrum and on the medial surface of the hind legs. These plaques typically remain small and are only of cosmetic concern. However, in rare cases, plaques can become extensive over the body or progress to neoplasia [4,5]. Canine viral plaques are caused by CPVs within the *ChiPV* genus [6].

Herein, the complete genome of a novel PV type is described. The PV type was amplified from a plaque that developed on the pinna of a dog [7]. The novel PV was designated CPV25, and classification as a species 3 *ChiPV* is indicated.

## 2. Materials and Methods

A 14-year-old spayed female West Highland White Terrier presented with a 2 cm slightly raised flat plaque on the concave surface of the right pinna close to the ear canal. The mass was surrounded by multiple similar small plaques, but no plaques were present elsewhere on the body. A sample of the largest plaque was excised, fixed in formalin, and submitted for histological examination. Histology revealed moderate hyperplasia of the epidermis. Additionally, many of the cells, especially in the granular layer, were enlarged with clear cytoplasm surrounding a shrunken dark nucleus. Smaller numbers of cells contained slightly granular blue cytoplasm. These cell features were consistent with PV-induced cell changes (Figure 1) Neither a scalloped appearance to the epidermis nor an accumulation of dermal melanin, both being features that are typically present within a canine pigmented plaque, were histologically visible. While the mass contained unusual histological lesions, it was classified as a viral plaque rather than as a papilloma [7].

A short section of a novel PV type was previously amplified from the lesion using the MY09/11 and CP4/5 consensus primers [7]. The MY09/11 amplicon was used to design primers to amplify the complete sequence of the novel CPV type. Briefly, to ensure that DNA was present within the lesion and not simply present as an incidental infection of the surrounding epidermis, a scalpel was used to take a sample from the center of the histology block of the viral plaque. DNA was extracted (NucleoSpin DNA FFPE XS kit, Macherey-Nagel, Düren, Germany), and PV DNA was amplified using the MY09/11 consensus primers, as previously described [5]. The positive control was DNA extracted from a lesion that contained CPV17, while no template DNA was added to the negative controls. The amplified DNA was sequenced and compared to other sequences in GenBank using BLAST (https://blast.ncbi.nlm.nih.gov/Blast.cgi; accessed on 1 February 2023).

A cotton bud moistened with saline was used to swab the affected area of the medial surface of the right pinna. DNA was then extracted from the swab using the same extraction kit as the fixed samples. PV DNA was amplified using the MY09/11 primers from the swab and sequenced. Outward facing primers were then designed using the sequence amplified using the MY09/11 primers. These primers allowed the circular nature of the PV DNA to be used to amplify the full genomic sequence. The primers (CPV25InvF 5′- TGAAGGGTTCATAGTGTGCA and CPV25InvR 5′-CGGTTTATAGATTCTGCGGC; Integrated DNA Technologies, Coralville, IA, USA) amplified an approximately 7500 bp region of DNA extracted from the swab of the pinnal plaque. The amplification was performed as previously described, and spades 3.15 and Geneious 10.2.6 were used to assemble the short sequences into a single sequence [8].

As the novel PV was most similar to CPV15 (Genbank number JX899359), the putative viral genes, *ORFs*, and regulatory sequences were predicted by comparing the novel sequence to this PV type, as previously described [8].

Complete genomes of 88 PV types from multiple genera were obtained from GenBank and aligned using MAFFT in Geneious v10.2.6 [9], and maximum likelihood analysis was performed, as previously described [3]. The sequence of the novel PV was deposited in GenBank under accession number OQ836189.

## 3. Results

Papillomaviral DNA was amplified from the swab of the plaques using the MY09/11 primers. These primers confirmed the presence of the same 357bp section of the *ORF L1* that was previously amplified from formalin-fixed samples of plaque [7].

The complete genome of the CPV was 7778bp, with a GC content of 52.2%. The first nucleotide in the *ORF* E6 was assigned number one in the sequence. As this is the 25th PV type detected in domestic dogs, it was designated CPV25.

CPV25 contained seven predicted *ORFs* coding for five early genes (E1, E2, E4, E6, E7) and two late genes (L1, L2; Figure 2). These results are shown in Table 1.

Due to the ability of the E6 and E7 proteins to influence cell growth and proliferation, these are typically the most important PV oncoproteins [6]. The putative CPV25 E6 protein contains 168 amino acids (aa) with two conserved zinc-binding domains (CXXC-X29-CXXC) between aa 47–83 and 120–156. There was no PDZ-binding motif (ETQL in the C-terminus of E6. A conserved zinc-binding domain was present between aa 53 and 89 within the 103 aa putative CPV25 E7 protein. The E7 protein was also predicted to contain a retinoblastoma (pRb) protein-binding site (LXCXE) at aa 23–27.

Viral replication and transcription is dependent on the E1 and E2 proteins. For CPV25 a E1 protein was predicted to be 626 aa in length. The predicted protein had N-terminal (aa 7–118) and C-terminal (aa 330–617) ATP-dependent helicase domains. An ATP-binding site (GPPNTGKS) was predicted at aa 454–461. PV DNA replication is dependent on cyclin/cyclin-dependent kinase complexes binding to the E1 protein [10], and two such cyclin A interaction sites (RXL) were detected in CPV25 at aa 106–108 and 554–556. At 524 aa in length, CPV25 also contained a putative E2 protein. A N-terminal transactivation helicase domain (aa 1–174) and a leucine–zipper domain (LX6LX6LX6L) at aa 440–461 were predicted in this protein.

The CPV25 genome was also found to contain a putative *ORF E4.* As in other PVs, this finding was identified in a different translation frame to the *ORF E2* region. The CPV25 E4 protein has two proline-rich domains, but does not have a high proline content (15.4%).

The viral capsid proteins are coded by the L1 and L2 proteins. CPV25 was found to have both *ORF L1* and *ORF L2* at 498 aa and 524 aa, respectively. As in other PVs, both L1 and L2 of CPV25 had many positively charged residues (K and R) in the C-terminal end, which are important in gaining entry into a host cell. The L1 protein had a Y-R dipeptide motif at aa 419–421, while the L2 protein had a furin cleavage motif (RXK/R-R) but no C-terminus L1-binding site at aa 488–491 (PXXP motif).

The CPV25 long-control region (LCR) is 422 bp in length (nt 7357–7778) between L1 and E6. While not coding for a protein, this region regulates transcription of viral genes. The CPV25 LCR contains four putative E2 binding sites, with a consensus sequence ACCN6GGT, at nt 7448, 7484, 7614, and 7691. No E1 binding site was identified.

The maximum likelihood tree comparing CPV25 with other PV types demonstrated grouping within the *ChiPV* genus (Figure 3). CPV25 clustered with the species 3 *ChiPVs* (CPV15, CPV10, CPV8 and CPV14) with high branch support. The *ORF L1* nucleotide sequence of CPV25 is most similar to CPV15, with 83.9% similarity to this PV type (Table 2). The CPV25 *ORF L1* sequence also has high similarity with other species 3 *ChiPVs*, including CPV8 (70.8%), CPV10 (69.5%) and CPV14 (67.8%). In comparison, CPV25 is less similar to the species 1 (CPV12 most similar with 62.3% similarity) and species 2 *ChiPVs* (CPV4 most similar with 61.1% similarity). When compared to CPV types of other genera, CPV25 has 55.5% similarity to the LambdaPV CPV1 and 54.5% similarity to the TauPV CPV2. Comparison of CPV25 with PVs from other host species revealed the most similarity with *Felis catus* papillomavirus type 2 (59.1%) and *Sus scrofa* papillomavirus type 1 (59.1%).

## 4. Discussion

When the CPV25 *ORF L1* was compared to those of other PV types, the only PVs that had greater than 60% similarity were all within the *ChiPV* genus. As PVs that share over 60% similarity in the *ORF L1* are usually within the same genus, this strongly suggests classification of CPV25 within the *ChiPV* genus. Furthermore, as the CPV25 *ORF L1* had the greatest similarity to the CPVs within species 3, it is most likely that CPV25 is the fifth species 3 *ChiPV* recognized to infect dogs.

The detection of CPV25 DNA in the viral plaque could indicate that the plaque was caused by the PV. However, due to the frequency with which PVs are present as asymptomatic infections, simply detecting the PV does not prove that the PV caused lesion development [11]. In the present case, the DNA that contained CPV25 was extracted from the center of the viral plaque. While this finding confirms that CPV25 was present within the plaque, this does not prove that CPV25 was causative. Further evidence of a PV etiology in the present case was the presence of PV-induced cell changes within the plaque. Finally, canine viral plaques are known to be caused by *ChiPV* types [1]. As CPV25 is also a *ChiPV*, and as PVs within the same genus often result in similar clinical lesions [2], the detection of CPV25 within a plaque containing PV-induced cell changes provides strong evidence supporting this PV type as the cause of the viral plaque. However, when using consensus primers, the possibility that the lesion contained a second, undetected PV type cannot be definitively excluded.

Since the initial reports of pigmented plaques in 2004 [12], there have been numerous evaluations of these lesions for PV DNA. These studies resulted in the complete sequences of 14 different canine *ChiPVs* being identified. However, despite the high number of plaques evaluated, CPV25 was not previously detected. While previous studies could have used methods that did not detect CPV25, the lack of previous amplification of this PV type suggests that CPV25 may be a rare cause of viral plaques in dogs. The presently described plaques were also unusual as they were confined to the pinna of the dog. To the authors’ knowledge, plaques in this location were not previously reported in dogs. Different PV types are recognized to preferentially colonize different areas of the body [13]. Therefore, it is possible that CPV25 could preferentially infect the skin of the pinnae of dogs. If this was the case, it is possible that CPV25 could be a more frequent cause of plaques at this location, while remaining a very uncommon cause of plaques elsewhere on the body.

The plaques in the present case were also unusual as they showed histological features that are not generally present in plaques identified at more common locations on the body. These atypical features include greater epidermal hyperplasia, frequent PV-induced cell changes, and a lack of pigmentation. Although it is necessary to identify additional plaques that contain CPV25, evidence from this case suggests that plaques caused by CPV25 may have histological features that are different to those in plaques caused by other CPV types.

The complete PV sequence was derived from a swab of the plaques. This approach was necessary as the diagnostic biopsy sample of the plaque was fixed in formalin and no additional tissue was available from the case. Formalin fixation causes breaks in the DNA strand that prevent larger sections of DNA from being amplified. The detection of CPV25 DNA within the swab suggests a productive infection within the plaque, with viral particles being present within the superficial epidermis. The presence of viral replication within the plaque was also supported by the numerous PV-induced cell changes visible within the lesion. Such florid PV-induced cell changes are rarely present within canine viral plaques, suggesting a higher than normal PV replication in the CPV25-associated plaque. Evaluation of additional plaques containing CPV25 will allow us to determine if this PV type consistently stimulates greater viral replication than other canine *ChiPV* types.

While the pathogenesis of canine viral plaques is not fully resolved, they are hypothesized to develop due to an inability of the immune system to suppress replication by PV types that are normally asymptomatically present on the skin’s surface [1]. The presently reported dog had chronic otitis externa in the same ear that developed the pinnal plaque. This chronic infection could have resulted in local disruption of the skin defenses, allowing plaque development. Alternatively, if CPV25 infection is restricted to the ears of dogs, the dog could have more generalized immunosuppression, but only develop viral plaques due to CPV25 on the pinna.

Further research is required to determine how CPV25 infection alters cell regulation. However, CPV25 is predicted to have a E7 protein with a pRb binding site. As pRb inhibits cell replication, binding this protein could allow increased cell division within infected cells [14]. In addition, zinc-binding domains were identified in the CPV25 E6 protein that could influence cell proteins and promote cell replication. Canine pigmented plaques rarely progress to neoplasia. In the present case, the plaques remained present for over a year without showing evidence of neoplastic transformation. Therefore, current evidence suggests that CPV25, like other Chi-PVs, can influence cell regulation, but not in a way that strongly predisposes cells to neoplastic transformation.

## 5. Conclusions

CPV25 is a novel PV that was identified within a viral plaque of a dog. This PV is the 25th PV fully sequenced from dogs. The presently described plaques were unusual due to their location and their histological features. As CPV25 was not previously identified in a canine viral plaque, this finding suggests the possibility that this PV type has a restricted location on the body and, when CPV25 causes lesion development, the infection results in unusual histological features.

## Figures and Tables

**Figure 1 animals-13-01859-f001:**
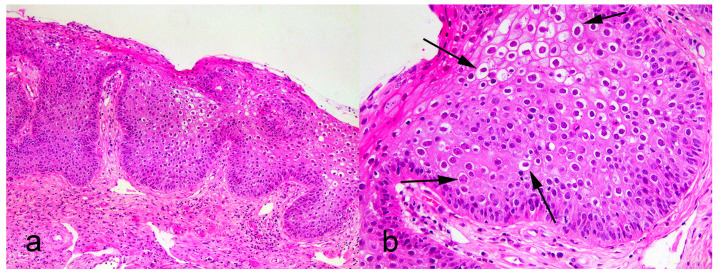
Photomicrographs of a plaque from pinna of a dog. (**a**) Epidermis is thickened, but not folded, as would be present within a papilloma H&E, 100×. (**b**) Thickened epidermis contains PV-induced cell changes, including enlarged cells with clear cytoplasm surrounding a shrunken dark nucleus (arrows) H&E, 400×.

**Figure 2 animals-13-01859-f002:**
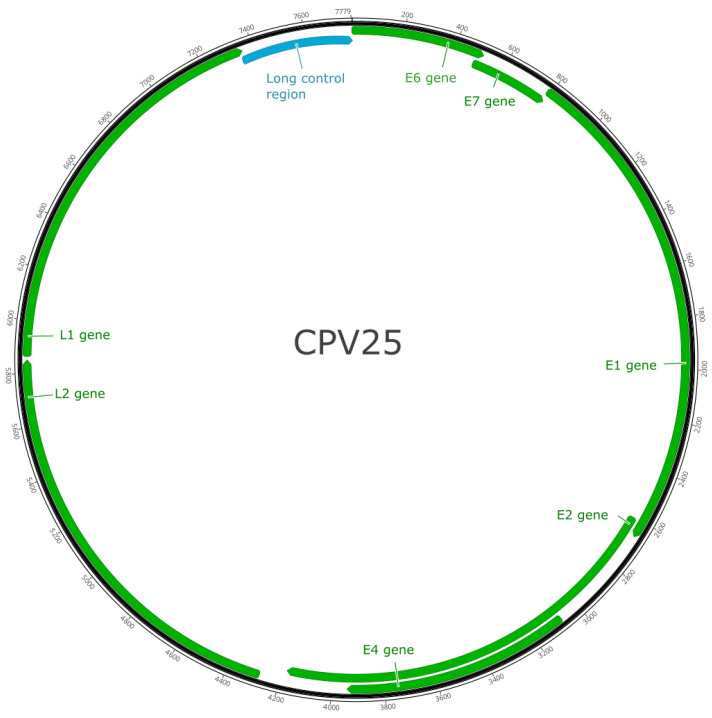
Schematic genomic organization of canine papillomavirus type 25.

**Figure 3 animals-13-01859-f003:**
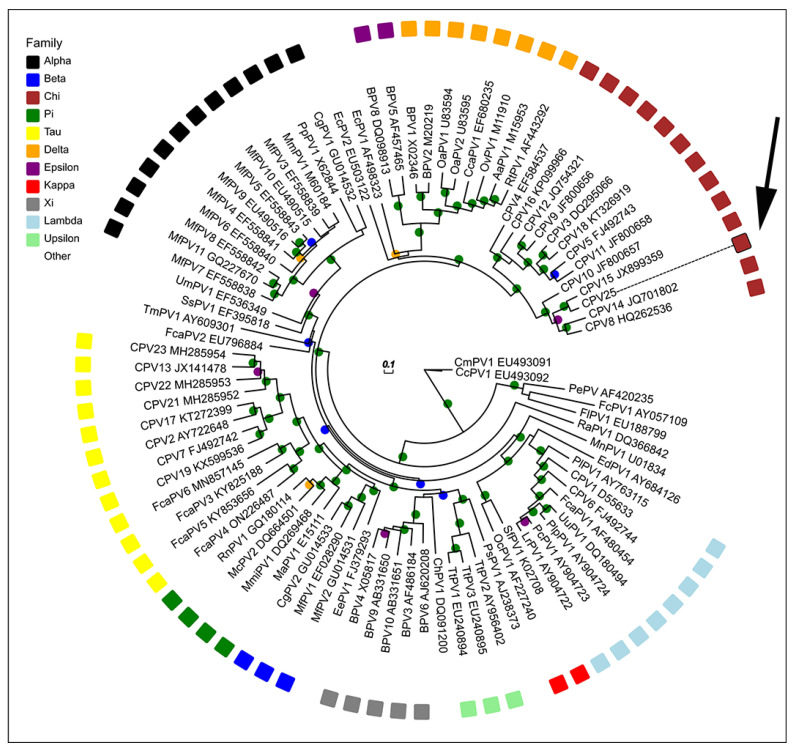
Unrooted maximum likelihood phylogeny based on concatenated nucleotide alignment of CPV25 *L1 ORF* sequence (arrow) with 88 other papillomavirus (PV) types of different species and genera. Accession numbers for sequences used are included. Abbreviations used include micromys minutus papillomavirus, MmiPV; mastomys coucha papillomavirus, McPV; canine papillomavirus, CPV; bovine papillomavirus, BPV; bettongia penicillata papillomavirus, BpPV; macaca fascicularis papillomavirus, MfPV; felis catus papillomavirus, FcaPV; equus caballus papillomavirus, EcPV; multimammate rat papillomavirus, MnPV; psittacus erithacus timneh papillomavirus, PePV; fringilla coelebs papillomavirus, FcPV; Francolinus leucoscepus papillomavirus, FlPV; Ovis aries papillomavirus, OaPV; oryctolagus cuniculus papillomavirus, OcPV; sylvilagus floridanus papillomavirus, SfPV; rousettus aegyptiacus papillomavirus, RaPV; capreolus capreolus papillomavirus, CcaPV; odocoileus virginianus papillomavirus, OvPV; alces alces papillomavirus, AaPV; rangifer tarandus papillomavirus, RtPV; erinaceus europaeus papillomavirus, EePV; colobus guereza papillomavirus, CgPV; mesocricetus auratus papillomavirus, MaPV; rattus norvegicus papillomavirus, RnPV, phocoena spinipinnis papillomavirus, PsPV; tursiops truncatus papillomavirus. TtPV; capra hircus papillomavirus, ChPV; trichechus manatus latirostris papillomavirus. TmPV; sus scrofa papillomavirus. SsPV; ursus maritimus papillomavirus, UmPV; macaca mulata papillomavirus. MmPV; pan paniscus papillomavirus, PpPV, erethizon dorsatum papillomavirus. EdPV; procyon lotor papillomavirus, PiPV; uncia uncia papillomavirus, UuPV; panthera leo persica papillomavirus, PlpPV; puma concolor papillomavirus. PcPV; lynx rufus papillomavirus, LrPV. PV genera are also listed. Internal branches are colored based on inferred bootstrap support values, as determined based on 1000 replicates using RAxML. Scale bar indicates genetic distance (nucleotide substitutions per site).

**Table 1 animals-13-01859-t001:** Predicted *ORFs* in CPV25 genome. pI indicates isoelectric point.

ORF	ORF Location	Length (nt)	Length (aa)	Molecular Mass (kDa)	pI
E1	783–2663	1881	626	71.18	5.53
E2	2605–4176	1572	524	56.63	7.36
E4	3119–3940	822	274	31.48	10.37
E6	1–507	507	168	19.08	6.72
E7	482–793	312	103	11.65	4.53
L1	5863–7356	1494	497	56.02	7.37
L2	4279–5850	1572	523	56.33	5.08

**Table 2 animals-13-01859-t002:** Percentage identity between proposed CPV25 and other papillomaviruses (PV) types. Alignments were performed using MAFFT in Geneious v10.2.6 using default parameters.

Papillomavirus	Host Species	Classification	L1 Similarity (%)
Canine familiaris papillomavirus 15 (JX899359)	Domestic dog	ChiPV3	83.9
Canine familiaris papillomavirus 8 (HQ262536)	Domestic dog	ChiPV3	70.8
Canine familiaris papillomavirus 10 (JF800657)	Domestic dog	ChiPV3	69.5
Canine familiaris papillomavirus 14 (JQ701802)	Domestic dog	ChiPV3	67.8
Canine familiaris papillomavirus 12 (JQ754321)	Domestic dog	ChiPV1	62.3
Canine familiaris papillomavirus 4 (EF584537)	Domestic dog	ChiPV2	61.1
Canine familiaris papillomavirus 1 (D55633)	Domestic dog	LambdaPV	55.5
Canine familiaris papillomavirus 2 (AY722648)	Domestic dog	TauPV	54.5
Felis catis papillomavirus 2 (EU796884)	Domestic cat	DyothetaPV1	59.1
Sus scrofa papillomavirus 1 (EF395818)	Domestic pig	DyodeltaPV	59.1

## Data Availability

The full sequence of CPV25 is available on GenBank.

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
