# Peer review of "Genomic Characterization of Canis Familiaris Papillomavirus Type 25, a Novel Papillomavirus Associated with a Viral Plaque from the Pinna of a Dog"

_animals, 2023, doi:10.3390/ani13111859_

Round 1
Reviewer 1 Report
This is a nice paper with elegant work on the genomic organization of the 25th canine PV. The authors have been careful not to over-reach with attributing causality, but have also done a good job of trying to provide evidence that this virus is an active participant in lesion development rather than a bystander. Their thoughts about location specificity of PVs are very interesting and I hope will lead to more cases of CPV25 disease being studied.
I have a few small questions to raise with the authors:
Lines 216-217: "While this confirms that CPV25 was within the plaque and not within the surrounding normal epidermis, this does not prove CPV25 217 was causative." How was it shown that CPV25 was not in the surrounding epidermis if only the centre of the lesion was dissected out from the FFPE block and tested? There is no mention of adjacent non-lesional skin being tested.
Manuscript as a whole: The authors used just one set of consensus primers (MY09/11), which detected the virus that turned out to be CPV25. Are they confident that this consensus primer set would have detected all 24 of the other known CPVs, which would let them know that CPV25 was definitely the only CPV in the plaque? In lines 227-228 they say, "However, despite the high number of plaques evaluated, CPV25 has not been previously 227 detected." Would CPV have been detected in all previous studies using the primers that each study's authors used (i.e., did all previous studies also use MY09/11?)?
There are just a few small typos or other suggestions, in no special order:
Lines 18, 25, 63 and others: The medial surface of the pinna is more correctly termed the concave surface, since "medial" has opposite meaningsx in breeds with pendulous vs. erect ears.
Line 178: Sus scofa should be Sus scrofa
Lines 177 and 178: Please italicize Felis catus and Sus scrofa
Line 201: Duplicated full stop
Line 244: Prevents should be prevent
Line 245: Production infection should be productive infection
Line 255: Otitis external should be otitis externa
Table 2: Column alignment is out (probably just in the proof and will be fixed in the final)
If one is available, a gross photograph of the ear lesion would be very helpful for clinicians, who may be unfamiliar with the appearance of pigmented viral plaques in this unusual location.
Lines 236-238: "The plaques in the present case were also unusual as they showed atypical histological features including greater epidermal hyperplasia, frequent PV-induced cell changes, and lack of pigmentation." It could be clearer to add "... when compared with pigmented viral plaques in more typical anatomic locations." or something like this.
Line 118: "Comparison of the 357bp section 117 of the ORF L1 revealed 100% similarity to a partial PV sequence previously submitted". Could you expand a little on this sequence? Which CPV did this belong to? Or was it banked as an unknown and then turned out to correlate with CPV25? Sorry, this was a little unclear to me.
Lines 67 and 68: "... frequent presence of PV-induced cell changes in keratinocytes within the granular layer. The epidermal spiking ..." It would be helpful to also describe these PV-induced changes here. Currently they're really only described in the legend for Figure 1. In addition, could you please clarify what epidermal spiking means? Is it the same as epidermal folding, which you use later in the manuscript?
Lines 67 and 68: It would be helpful to refer readers to Figure 1 in this sentence.
Author Response
This is a nice paper with elegant work on the genomic organization of the 25th canine PV. The authors have been careful not to over-reach with attributing causality, but have also done a good job of trying to provide evidence that this virus is an active participant in lesion development rather than a bystander. Their thoughts about location specificity of PVs are very interesting and I hope will lead to more cases of CPV25 disease being studied.
Thank you for the positive feedback.
I have a few small questions to raise with the authors:
Lines 216-217: "While this confirms that CPV25 was within the plaque and not within the surrounding normal epidermis, this does not prove CPV25 217 was causative." How was it shown that CPV25 was not in the surrounding epidermis if only the centre of the lesion was dissected out from the FFPE block and tested? There is no mention of adjacent non-lesional skin being tested.
The authors agree that this statement is incorrect and that we did not prove that CPV25 was not in surrounding epidermis as stated. This statements have been modified accordingly to reflect the testing that was actually performed in this case (line 208)
Manuscript as a whole: The authors used just one set of consensus primers (MY09/11), which detected the virus that turned out to be CPV25. Are they confident that this consensus primer set would have detected all 24 of the other known CPVs, which would let them know that CPV25 was definitely the only CPV in the plaque?
The authors apologise for not explaining clearer. In the case report describing these lesions (ref 7) it is described how both MY09/11 and CP4/5 consensus primers were used with both primers only amplifying a short section of DNA from what turned out to be CPV25. In the present report as only the product of the MY09/11 primers was used for complete sequencing, only this method is described. This additional information has been added, along with the possibility that other PVs could have been present in the lesions (line 76, 214)
In lines 227-228 they say, "However, despite the high number of plaques evaluated, CPV25 has not been previously 227 detected." Would CPV have been detected in all previous studies using the primers that each study's authors used (i.e., did all previous studies also use MY09/11?)?
This is an excellent point and a statement that it is uncertain whether or not the previous studies used methods that would have detected CPV25 has been added (line 221).
There are just a few small typos or other suggestions, in no special order:
Lines 18, 25, 63 and others: The medial surface of the pinna is more correctly termed the concave surface, since "medial" has opposite meaningsx in breeds with pendulous vs. erect ears.
Corrected as suggested (line 18, 26, 65).
Line 178: Sus scofa should be Sus scrofa
Corrected (line 170)
Lines 177 and 178: Please italicize Felis catus and Sus scrofa
The authors have previously been told by other reviewers that when describing a cat then Felis catus is correct but when describing the name of a papillomavirus Felis catus papillomavirus type 2 is correct. Looking through other publications it does appear most adhere to this and so this has not been changed within the manuscript.
Line 201: Duplicated full stop
Corrected.
Line 244: Prevents should be prevent
Corrected (line 239)
Line 245: Production infection should be productive infection
Corrected (line 240)
Line 255: Otitis external should be otitis externa
Corrected (line 250)
Table 2: Column alignment is out (probably just in the proof and will be fixed in the final)
Corrected
If one is available, a gross photograph of the ear lesion would be very helpful for clinicians, who may be unfamiliar with the appearance of pigmented viral plaques in this unusual location.
Unfortunately no photograph of the lesion is available for this manuscript.
Lines 236-238: "The plaques in the present case were also unusual as they showed atypical histological features including greater epidermal hyperplasia, frequent PV-induced cell changes, and lack of pigmentation." It could be clearer to add "... when compared with pigmented viral plaques in more typical anatomic locations." or something like this.
This sentence has been changed to improve clarity (line 231).
Line 118: "Comparison of the 357bp section 117 of the ORF L1 revealed 100% similarity to a partial PV sequence previously submitted". Could you expand a little on this sequence? Which CPV did this belong to? Or was it banked as an unknown and then turned out to correlate with CPV25? Sorry, this was a little unclear to me.
The authors agree that this was confusing and have changed the sentence to improve clarity (line 113)
Lines 67 and 68: "... frequent presence of PV-induced cell changes in keratinocytes within the granular layer. The epidermal spiking ..." It would be helpful to also describe these PV-induced changes here. Currently they're really only described in the legend for Figure 1. In addition, could you please clarify what epidermal spiking means? Is it the same as epidermal folding, which you use later in the manuscript?
The sentences in these lines have been improved as suggested (lines 68-74)
Lines 67 and 68: It would be helpful to refer readers to Figure 1 in this sentence.
This has been added as suggested (line 72).
Reviewer 2 Report
This article describes the discovery of a new strain of canine papilloma virus, isolated from a lesion on the pinna of a dog. The authors isolate the viral DNA and PCR amplified the full genome, which was sequenced and compared to other CPV deposited in the data bases. Although of some interest to general audience, the data is well described, but not well presented, there are figures that are not mentioned in the text (Figure 1) and the text describes figures that do not exist (figure 2A). The presentation of the data must be improved if the manuscript is to be considered for publication.
Specific observations:
1. Line 33 abstract “This PV type may a rare cause..” May cause a rare?
2. Figure 1 need a lot of improvement. A and B are kind of lost due to the color use. There is no other label in the figure which areas are been shown? where is the lesion? Which ones are the cells mentioned in the legend? Also, the figure is not cited in the text.
3. Figure 2 does not have a 2a part as indicated in the text
4. Figure 2 only shows the different ORFs, the authors described the identification of a regulatory sequences, this should also be depicted in the figure.
5. Line 173 has repeated words
Author Response
This article describes the discovery of a new strain of canine papilloma virus, isolated from a lesion on the pinna of a dog. The authors isolate the viral DNA and PCR amplified the full genome, which was sequenced and compared to other CPV deposited in the data bases. Although of some interest to general audience, the data is well described, but not well presented, there are figures that are not mentioned in the text (Figure 1) and the text describes figures that do not exist (figure 2A). The presentation of the data must be improved if the manuscript is to be considered for publication.
The authors apologize for the sloppy presentation of data. This has been improved within the revised manuscript.
Specific observations:
- Line 33 abstract “This PV type may a rare cause..” May cause a rare?
This sentence has been revised to improve clarity (line 22)
- Figure 1 need a lot of improvement. A and B are kind of lost due to the color use. There is no other label in the figure which areas are been shown? where is the lesion? Which ones are the cells mentioned in the legend? Also, the figure is not cited in the text.
The authors agree that this figure needed improvement. A suggested by the reviewer, the A and B has been enlarged and thickened to make them easier to see. The color balance of the figure has been improved to make it easier to see features. The figure legend has been improved and arrows have been added to make it clear which cells are referred to in the legend. The figure has now been cited in the text (line 116).
- Figure 2 does not have a 2a part as indicated in the text
This has been corrected (line 124)
- Figure 2 only shows the different ORFs, the authors described the identification of a regulatory sequences, this should also be depicted in the figure.
This has been added as suggested.
- Line 173 has repeated words
This has been corrected (line 162).
Round 2
Reviewer 2 Report
The authors have improved the presentation of the paper, and unswer all my questions.